# The Evolution of University Students' Financial Attitudes and Their Role in the Sustainability of Personal Finances

**Zoltán Zéman** [1], **Botond Géza Kálmán** [2,*] , **Judit Bárczi** [1] and **László Pataki** [1]

[1] Faculty of Economics and Business, John von Neumann University, 6000 Kecskemét, Hungary; zeman.zoltan@uni-neumann.hu (Z.Z.); barczi.judit@uni-neumann.hu (J.B.); pataki.laszlo@uni-neumann.hu (L.P.)

[2] Institute of Economics, Faculty of Business, Communication and Tourism, Budapest Metropolitan University of Applied Sciences (MET/ÜKT GTI), 1148 Budapest, Hungary

[*] Correspondence: eupemq@instructor.metropolitan.hu; Tel.:+36-305273344

**Abstract:** The purpose of this study is to examine the question of how crises influence the decision-making of Hungarian university students. Crises increase the risk of sustainability, so it is crucial to make appropriate financial decisions in such a situation. For this purpose, the authors conducted a two-stage questionnaire survey among students of economics and other majors. The inquiries took place in 2019 ($n = 1558$) and 2020 ($n = 1712$). A regression study was used to analyse the evolution of financial attitudes and investment knowledge, as well as how they are affected by a potential crisis modelled with the COVID-19 pandemic. It has been shown that interest in financial matters increases as a result of the crisis and the level of financial knowledge also increases. However, the most important conclusion of the study is that, in the event of a high threat, knowledge and practice can only be combined with calm thinking to help make appropriate financial decisions. All of this together ensures that investment decisions are the basis for the sustainability of personal finances.

**Keywords:** financial decision; economic crisis; university students; sustainable personal finances

## 1. Introduction

Is it common sense what the following terms mean: factoring, Lombard credit, revolving credit, letter of credit? Is there an unequivocal answer for that? Then, perhaps, have you heard of Revolut or Transferwise? At the very last, have you even got an idea about what might lie beneath their trading name? It is not crucial to have this knowledge since if you need such products, you can look them up. But if you even have difficulty knowing what interest, maturity, and instalments are, you are hundreds of years behind in basic financial knowledge. Nowadays, loans–and financial services in general–have become part of our lives. Especially in crises, when jobs and incomes may be at risk. Although taking out a loan is not the right solution to alleviate the daily financial problems, we need to understand this, we need at least basic financial knowledge. And these do not come into our heads by themselves, they have to be learned.

The acquisition of inanceal knowledge is effective if the appropriate content is delivered by the appropriate specialist using the necessary methods. Based on these, it is almost self-evident that the frameworks of organized education, public education and higher education are ideal for the realization of this goal. Based on theoretical considerations, it is most appropriate to teach financial knowledge as part of the uniform high school curriculum. In 2017, the Government of Hungary launched a national strategy aimed at developing financial awareness [1]. Part of this is the teaching of financial knowledge at an appropriate level in primary and secondary schools. The impact of this program will only be felt from 2024–2025. Until then, however, the financial education of the current generation of university students must also be ensured. Given that university students belong to the generation that is just about to enter the labour market, there is already a stake

in learning finance for them. Their acquired knowledge will be evaluated by a practical test within the foreseeable future. This study presents the financial culture of university students, focusing on a few special areas: the relationship (attitude) to financial matters, and a distinguished group of financial knowledge, namely, investment knowledge. We chose the latter because they are one of the possible elements of the personal financial safety net in a crisis.

The economic crisis of 2008 and Its effects on everyday people focused the attention of specialists on several key areas. Such an area is, for example, financial literacy, which, in addition to the theoretical knowledge necessary for proper and efficient management of everyday finances, also includes practical competencies [2]. This includes, among other things, the investment decisions that are the subject of this study, as well as the theory and practice related to insurance related to them. Nowadays, the importance of financial knowledge and competencies is becoming more and more important. The previous dominance of cash in everyday transactions is gradually being replaced by card and online payments, and new financial services, insurance and investment schemes are spreading. Their proper selection and management requires more and more knowledge. The same abilities and skills are needed to create a sense of financial security. One of the pillars of this security is savings that establish the future [3], which can be a source of unforeseen and unplanned expenses. The index of financial security [4] primarily calculates available, liquid or easily convertible financial resources. At the same time, savings for retirement are important indicators of planning for the distant future, and there is also a correlation between the ability to plan long-term and university studies: in the group of students who finance their university studies with student loans, later savings for retirement are 36 percent less than in the case of self-employed students [5]. Established financial security becomes a particularly important factor in the event of environmental crises and economic crises. In such situations, the role of education comes to the fore, as good decisions need to be made quickly [6] Decisions made in a crisis are primarily determined by the amount of previous emergency savings the person concerned has. According to Treger and Wendel [5], these savings have a much stronger impact on financial decisions than income levels. The financial crisis of 2008 drew attention to the fact that incorrect personal financial decisions made during the crisis necessitate state intervention. The failure of investments can even lead to a crisis in the financial institution system. And this can divert considerable budgetary resources from the original goals, endangering the already critical sustainability of the economic situation.

The basic premise offer line of thought leading to our topic is that we live today, in the era of Industry 4.0. One of its characteristics is the exponentially accelerating scientific and technical development. This is necessary, since the limited resources of our Earth must be used to take care of an ever-faster growing population. It is no coincidence that sustainability is a common concept in the literature, whether it is economics [7] or education [8]. The basic condition for ensuring sustainability is economic stability. At the macro level, this means preventing economic recessions. However, it is also necessary to ensure individual financial stability. This requires adequate financial knowledge. Since the national strategy aimed at developing financial awareness in Hungary was launched in 2017' today's generation of university students can only get a chance to acquire the appropriate knowledge within the framework of higher education. One of the pillars of universities' competitiveness and thus their sustainability can therefore be the comprehensive education of economic knowledge for all students. Due to the increasingly close cooperation between industry and higher education, competitiveness has become important not only for companies but also for universities [9,10]. One possible pillar of this is the attraction of quality-centred education, which attracts students and, with them, resources to universities. And these are also the basis of the sustainability of education.

Intensive research on the financial literacy of university students began in the 1990s [11,12]. These early studies highlighted the importance of financial education in the development of financial culture. Later, it was also proven that the financial culture of the students of economic

courses is different [13]. Two landmark studies were also published in 2010 that established the methodology for measuring financial literacy [14,15]. In recent decades, an increase in the value of financial knowledge can be observed. The reason for this is that online or digital payments are increasingly taking the place of cash transactions, and new credit, investment and insurance schemes are constantly appearing. Managing them definitely requires increasingly complex financial knowledge. In addition to knowledge, the Organization for Economic Co-operation and Development (OECD) also examines financial behaviour and attitudes in its financial culture surveys [16].

Already at the beginning of the 2000s, it was shown [17] that teaching investment skills can significantly increase the financial knowledge of the subjects who participate in them. The importance of the topic of investments is also underlined by the fact that these are often the financial activities with the greatest risk, just think of option transactions or real estate funds. It is not by chance that the concept of casino capitalism was born for this economic area [18]. On the other hand, this is the area where a very significant flow of money takes place without actual production or service provision, hence the concept of the paper economy [19]. At the same time, investments are usually indicators of the ability to plan long-term. However, my current topic is related to the issue of economic crises, which we modelled with the current COVID-19 pandemic. Nevertheless, the role of investments is still important even in crises. They are part of the personal financial safety net that is the key to financial security in a crisis.

Financial literacy is a component of financial attitude, i.e., an individual's relationship to money and financial matters [20], which is a motivational factor for financial behaviour. Neither financial knowledge nor parents' income has a significant effect on financial behaviour, but financial attitude does [21]. Financial education should be based not only on the lexical knowledge deemed important by the organizers, but also on the students' opinion and attitude towards money. Since young people of a similar age differ significantly in their financial attitudes, an effective financial education program must also meet the training needs of students with different financial skills, knowledge and abilities. The impact of high school and university financial education can best be seen on former students. One possibility is to examine the extent to which adult investment decisions are correlated with majors in high school and university studies [22]. According to the results, high school course participation is not at all related to adult investment decisions, and even in the case of university education, the decisions made were more influenced by one's own financial experiences than studies.

Other authors [23] also found that in relation to practical knowledge related to investments, students of economic courses perform better than their peers attending different courses, in the following way: 5–35 percentage points advantage in everyday questions, 5–10 in the field of insurance, and 40–80 in the field of investment knowledge. Other research also reached the same result [16]. At the same time, it is also a proven fact that the knowledge acquired through appropriate training also helps in making investment decisions [24]. It is especially important to be aware of this, given that financial awareness and the setting of long-term financial goals are less common among young adults [25]. The Money Attitude Scale [26] has been used to measure financial attitudes since the early 1980s. Several domestic attitude measurements were carried out [27–29]. These researches examined young people in different demographic distributions. They managed to prove a parallel between childhood traits and adult attitudes towards finances. One form of attitude research is based on personality typing.

At the beginning of the 2000s, financial attitude research unanimously concluded that students tend to take out loans and use credit cards, which is partly a consequence of the higher bank lending willingness associated with economic growth. All of this is also related to the consumer behaviour of the university generation [11,12]. Covering his daily consumption needs (food and drink, entertainment) with a credit card was also no stranger to this approach. One of the bases for this was the conviction that he would definitely be

able to pay off the card debt on time, and the other was the widespread belief that spending the bank's money on the credit card is actually an income supplement [30].

The OECD [16] uses a different approach and measures the financial attitude using a method based on practical behavioural aspects that completely disregards personality traits [15] Kossev, whose methodology examines the ability to plan for the long term as a single key factor, scoring it on a five-point scale. Both financial attitude and financial skills are related to financial behaviour. Members of the college generation with better financial attitudes and skills demonstrate good financial behaviour in managing their money [31]. These results are significant and support the results of other studies [32], which established a close relationship between financial attitude and financial behaviour. The findings of the studies cited above are also significant and consistent with the finding that financial behaviour is predicted by financial attitude, but financial knowledge is not. However, we argue that increasing financial literacy is an important factor in promoting good financial decision-making.

Reviewing the literature on the topic and our own experience in university education, the authors found that there were few surveys on the subject in the geographical region of Central Europe, nor were these studies covering several countries. Therefore, university students from three countries (Austria, Hungary, Slovakia) were asked in their survey. The past of the three countries was still common at the beginning of the 20th century, they were all part of the Austro-Hungarian Monarchy. However, after the Second World War, Austria became part of Western Europe, while the Slovaks and Hungarians fell into the Soviet sphere of influence. Currently, all three countries are members of the European Union, Austria and Slovakia are also members of the eurozone. We also analyzed the appearance of these differences in economic education in the original research, of which this study describes only a small part. Based on the above, we formulated the following research hypotheses.

**H1.** *University students studying economics perform better in the area of financial literacy than their peers studying other subjects, whose education only includes economic knowledge.*

**H2.** *Those university students who undertake work in addition to their studies are more efficient and successful in solving financial problems, as practical experience is added to their theoretical knowledge.*

**H3.** *Those students who are only studying and do not take on a job in addition try to compensate for the lack of practical experience with a stronger financial attitude than students who work alongside the university.*

## 2. Materials and Methods

In our current research, we focused on students in higher education. They are the ones who will soon appear on the labour market with a diploma in hand, where they will have to apply what they have learned in practice. Our goal was to investigate whether there is a difference between the investment knowledge and financial attitude of economics and other university students. In our study, we focus on two effects that can shape both student knowledge and attitude. One of these is the work done in addition to the studies, the other is a possible crisis, which we modelled in our study with the pandemic caused by COVID-19.

As a sampling procedure, we chose a method with which we can expect to obtain a relatively large sample quickly and with little financial expenditure, but at the same time it can be considered scientifically accepted. Taking the above aspects into account, we decided on time-space sampling. It was originally developed for the purpose of researching rare (hidden) populations [33]. It is characteristic that the researcher questions such populations by visiting the places they like and visit at suitable times, because they can be reached here in a concentrated manner in space and time. Although university students do not hide, there are also places and times when we can find them concentrated. This is the university,

in education time. Acquaintances helped us in answering the questionnaires. They not only answered our questions, but also took them to their university, asked their fellow students to fill them out voluntarily and anonymously, and then returned the answered questionnaires to us. In other words, we used the respondent-driven version of spatial-temporal sampling. With this method, as we had hoped, we managed to collect a significant questionnaire in a short time.

Of course, the procedure is not without problems from a statistical point of view. One questionable feature is that the interrogators cannot visit all locations. Therefore, on the one hand, the entire population cannot be assessed, and on the other hand, the sample will almost certainly not be representative, and this difficulty cannot be remedied by subsequent weighting. Comparing the procedure we used with the results of cross-sectional studies [34], it can be concluded that the results obtained in this way should be treated with reservations. At the same time, by appropriately increasing the sample size, based on the Central Limit Theorem (CLT) [35], it is possible to collect a sample characterized by normality [36]. That is, considering the sample size, the results that can be evaluated with the procedure are obtained, which can be analysed within the limitations of the method and conclusions can be drawn.

The Instrument of our research was an offline questionnaire composed of self-made questions. In choosing this method, we were motivated on the one hand by mitigating the distorting effect and on the other hand by the hope of a higher response rate [37,38]. Our idea turned out to be correct in the end. We managed to get a high response rate of 92 percent, compared to the average rate of 30 percent for online questionnaires. Our questionnaire consisted of several groups of questions, we only deal with certain highlighted part of them (with 5 questions about investment knowledge) in this work, as the questions were compiled for the purpose of an ongoing, multi-country research. Our present study is based on the already processed partial results of this research. Financial culture was characterized by the triad of financial knowledge–financial behavior–financial attitude [12] (Kossev). The questions and statements regarding these were created after studying the literature [12,39].

Demographic questions enabling the description of the sample were included in the first group of questions. According to the literature [40], most of these demographic characteristics play a role in the development of financial behaviour. We also examined their relationship with financial attitudes. Demographic characteristics were interpreted in a wider than usual scope. That is, the questions included not only the respondent's gender, age, type of residence, and relationship status. In addition to the higher education institution and field of study visited by the respondent, we placed in this block also the questions related to the employment relationship at the university (full-time or correspondence curriculum, physical or intellectual work, manager or subordinate). The other groups of questions measured the following characteristics: financial knowledge, financial behaviour, financial attitude, feeling of financial security, financial preferences, student stress. In this research, the authors used the values of the attitude index created during their previous research from these groups of questions when creating their regression model. The value of the index was calculated by aggregating the scores given by the respondents to the attitude questions of the entire questionnaire.

The investment knowledge analysed in the current work covers the area of securities, insurance and pensions, supplemented by knowledge of the risks of investments. We examined five questions of our full questionnaire which are related to this area.

Q1.  The yield of life insurance combined with an investment always covers the amount to be repaid
Q2.  Do you think it is worth taking out insurance for investment purposes?
Q3.  Money market funds have no risk as our money is only deposited in bank accounts and invested in securities;
Q4.  There are also risk-free investments;

Q5. A part of our pension contribution goes to our registered account at the voluntary pension fund.

We treated Q4 as a priority question, because the answer to this alone shows whether the respondent is aware of the necessary knowledge at a basic level. The question was asked as follows: To what extent do you think the following statement is true: "There are also risk-free investments"? For the answer, we used a Likert scale, homogenized only at the end points. A value of 1 was not true at all, and a value of 5 was completely true. We did not label options 2, 3, 4, here we left it up to the respondents to place themselves in the continuum between the two endpoints. We evaluated answer 1 as a good answer, all other marks were classified as incorrect answers. We should all be aware of Milton Friedman's statement that "There is no such thing as a free lunch" [41]. At their age, university students can be expected to be aware of the fact that all investments involve risk and that risk is greater the higher the promised return. Even an investment considered as safe as a government bond has risks [42]. Therefore, those respondents who believe that there is a risk-free investment do not even have the most basic investment knowledge. This fact is independent of how they answered the several questions on the subject of investments.

All the questions in our questionnaire (except for age) were closed, partly to be decided (yes/no), partly answerable on a multi-point Likert scale. Investment knowledge was measured using a five-point Likert scale. The Likert scale was prepared by coding only the endpoints with strongly agree and strongly disagree definitions. No labels were made for the additional scalepoints. This procedure is scientifically accepted [43,44]. Its advantage is that it allows for a more differentiated response and leaves it up to the respondent to place themselves on the five-point scale. We created our own indices by aggregating the scores given by the respondents [45,46]. Our questionnaire was finalized using a pilot test before the actual data collection began. The students who took part in it also felt that the questions about the income category and the existence or non-existence of loans to be paid were excessively personal. Therefore, although the range of conclusions that can be drawn would have been significantly broadened from the point of view of our investigation, these questions were not included in the final questionnaire, on the one hand, for reasons of research ethics, and on the other hand, in order to achieve a higher response rate.

The surveys were conducted twice, first in September 2019 and second in September 2020, during the attendance education period. The purpose of the research, originally planned for one phase, was later changed due to the 2020 pandemic, the first phase of which was completed by September 2020, and we were in a period of subsequent easing. The university students once again took part in face-to-face education. They had time and opportunity to evaluate the experiences of the first wave. We therefore repeated and extended our study to assess the effects of the crisis. In both years, the students of the same five Hungarian universities answered:

- Budapest Business School (Budapest, Hungary)
- Budapest Metropolitan University (Budapest, Hungary)
- Eötvös Loránd University (Budapest, Hungary)
- Eszterházy Károly Catholic University (Eger, Hungary)
- University of Pécs (Pécs, Hungary).

In both years, the number of respondents in all student groups significantly exceeded the one hundred people usually mentioned as the automatic eligibility criteria for the CLT (and can be considered as a significant over insurance) [35] (Figure 1). The answers were processed using IBM-SPSS Statistics, IBM-SPSS-Amos and the R program package. Based on the processed results, we first decided to group the individual specializations into three larger groups based on the performance of the students attending them. Based on the performance values measured for the respondents, we modified the originally planned grouping (economics students-others). Using the regrouping method [47], we created three groups (economic-legal-other). The performance of law students turned out to be very similar to that of economics majors. By the way, this has already been reported by other authors [48]. One is for economics students, the other is for law students, and the third

group is a mix of students from all other majors, typically pedagogy, humanities and arts students. Highlighting the law students as a separate group was justified by the fact that their results were very similar to the performance of economics students. Interestingly, this was also observed in research in other literature [48].

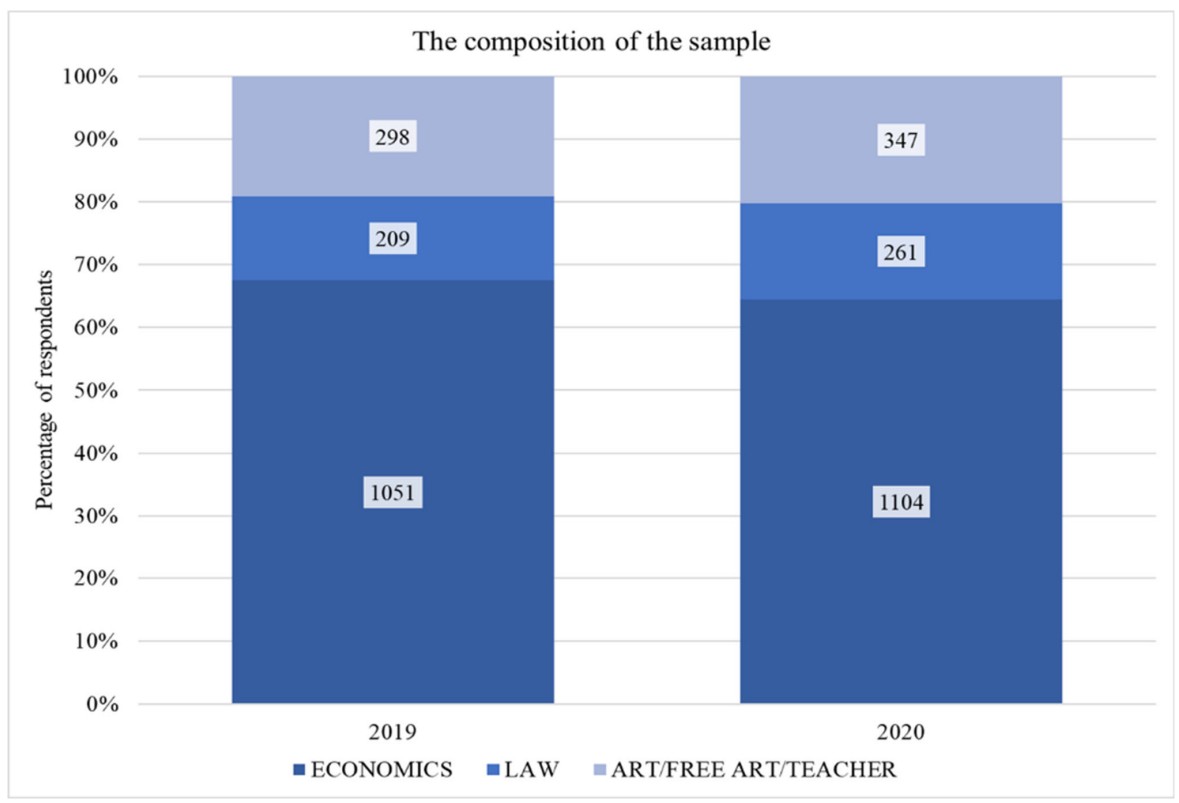

**Figure 1.** The structure of the sample.

Indices were created from the responses to each group of questions. We used aggregation as a method. We chose this method because it is simple and also considered an established method in the social sciences and psychology [45,46]. This can basically be done in two ways: using averaging or factoring. In an effort not to unnecessarily complicate the methodology, we chose averaging. Given that we did not give five-level answers to all of our questions, we defined and created our index as a number between 0 and 1 in order to make the index values comparable. In the case of questions where yes or no answers were possible, a "correct" answer corresponding to the index was worth one point, and an "incorrect" one not corresponding to it was worth zero point. If there were more than just two options for the answer, then we transformed the multiple answer options so that the least suitable index was zero, the most suitable was one, and the rest took a proportional place between the two extreme values. Finally, we came up with the index by averaging these values.

The effect of the exercise on the financial attitude, taking into account the normality of the sample, was investigated using the analysis of covariance (ANCOVA) method. Due to the high number of elements, the normality of the sample was taken as a given based on the CLT and no separate normality test was performed. After establishing normality, we examined the homoscedasticity of the sample. For this purpose, we used Levene's test, the results of which can be seen in Table 1. Since according to the null hypothesis o' Levene's test the variances are the same, homoscedasticity can be established based on our results. Our next step is to select the test to be used. Given that we have to compare more than two groups, and that our study focused solely on the effect of the Year of survey and Employment status variables, the ANCOVA procedure proved to be appropriate. With

this method, it is possible to filter out the influence of variables that play a role in the development of differences, but are not relevant for us in this research, such as the variable that shows that the respondent is studying economics, law or other training. Another condition for ANCOVA to be performed is that the dependent variable shall be continuous and the independent ones shall be categorical. Based on the description of the variables, this condition is also met. Given that the ANCOVA itself only shows the existence of a significant difference, we used the Tukey test as a post-hoc test to examine which groups this significance can be measured between. Among the differences found, the information that can be interpreted in this research is presented and analyzed in the description of the results. The significance level was uniformly set at 5 percent. The starting model was the following:

$$Y = \beta_0 + \beta_1 X_{1i} + \beta_2 X_{2i} + \beta_3 X_{1i} X_{2i} + e_i \qquad (1)$$

where:

$X_1$—Year of survey (2019; 2020)
$X_2$—Employment status (Physical worker, and university student, Intellectual worker, and university student, Only university student).

**Table 1.** Descriptive and homogeneity statistics of financial attitude (Source: Authors' own elaboration).

| Sample | N | Minimum | Maximum | Average | St. Dev. |
|---|---|---|---|---|---|
| Financial Attitude | 3270 | 0.13 | 1.00 | 0.72 | 0.16 |
| Homogeneity test | F | df1 | df2 | p | |
| Levene's | 0.158 | 1 | 3260 | 0.691 | |

The dependent variable in the model is Financial attitude. This variable shows how much an individual is attracted to finance. The concept of attraction can be interpreted both theoretically and practically. The theoretical attitude shows how much an individual is interested in financial information, and the practical attitude shows how willingly they manage their finances. The value of the Financial attitude variable was determined by averaging the value of the true-false answers given to the five selected questions of the questionnaire. Thus, in addition to the nominal (true-false/1-0) independent variable, we obtained a continuous dependent variable. We used two variables as independent variables. One is Year of survey. the most important change in the year between the two surveys was the emergence of the COVID-19 pandemic and the resulting global economic crisis. Therefore, we modelled and examined the impact of the economic crisis on financial attitudes with the Year of survey variable. Since we performed two queries, the variable is categorical and can take only two values: 2019 or 2020. Another independent variable is Employment status. This shows whether the theoretical knowledge obtained by the respondent from the university is complemented by the practical skills acquired on the labour market. This shows whether or not the respondent is committed to their university studies. With this variable, we examined the role and effect of practice in financial attitudes, as we assumed that practical experience complements theoretical knowledge and increases the level of financial literacy. For a more differentiated analysis, we separated blue collar and white collar work. Thus, we also obtained a categorical variable with three values (student and manual worker; student and intellectual worker; student only). A student and manual worker is a university student who undertakes physical work in addition to his studies, and a student and intellectual worker is a student who performs intellectual work in addition to his studies. A student only is a student who does not work while completing his university studies.

We asked our questionnaire in two phases among the students of the examined universities. In the first stage, a total of 1558 respondents filled out evaluable questionnaires, and in 2020, 1712 people participated in the survey. It would be tempting to explain the increase with increased interest due to the consequences of the coronavirus pandemic that broke out in the meantime, but a number of other reasons could also have contributed to

this, for example, the fact that our survey contacts were more successful in getting students involved in the study. Figure 1 shows the distribution of the sample according to university majors during the individual queries. The female-to-male ratio was close to 40–60 percent in both years. Most of the students included in the sample are only pursuing university studies and do not work alongside it. There were 1217 (78.1%) of them in the samples in 2019 and 1363 (80.2%) in 2020 (Figure 2). Students who work in addition to their university studies are mostly involved in intellectual activities, so they were represented in a larger proportion in the sample: 259 people (16.7%) in 2019, 260 people (15.1%) in 2020. Roughly every twentieth student does physical work in addition to their studies: 82 people (5.2%) in 2019, and 89 people (5.0%) in 2020.

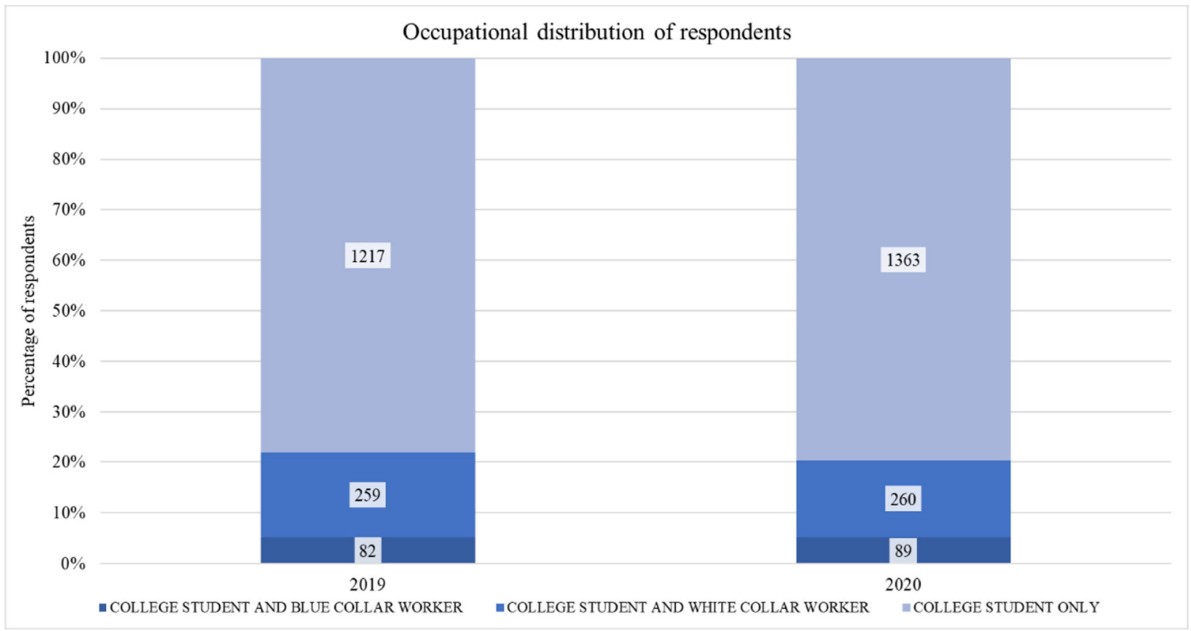

**Figure 2.** Occupational distribution of respondents (Source: Authors' own elaboration).

## 3. Results

### 3.1. Financial Attitude

The descriptive statistics of the financial attitude are included in Table 1. What we should observe in this regard is the average, which has a value of 0.72 (in other words, 72 percent). This indicates that students in higher education have a significantly better relationship with finances than the average. According to the OECD's latest survey [16], the entire Hungarian population achieved a score of 3.3 on the five-point attitude scale, which corresponds to a score of 66 percent. The financial attitude of higher education students is therefore not only significantly higher than the theoretical average (0.5), but also 6 percentage points better than the average of the entire Hungarian population. The attitude of university students is the same as that of students in other countries. The attitude score also has a value of 0.72 on a scale between 0–1 in the study conducted by Setiyani and Solichatun [49] in 2019 among Turkish university students. Swiecka et al. [50] in 2020 measured a value of 0.70 when examining Polish university students.

The regression model of financial attitude is significant, but has relatively low explanatory power. Among the parameters included in the model, the year of the survey and the status of the student have a significant effect, however, the interaction of these two variables is not. Among the two significant effects, the explanatory power of status is the largest ($\eta^2 = 0.0219$), its effect size is weak-medium. However, the size of the effect of the year is very weak ($\eta^2 = 0.0029$). Based on the results, the first year of the COVID-19 crisis did not significantly affect students' financial attitudes, but the impact on attitudes was still significant. Since a crisis always forces change, we assume that the effect of the year

of the survey was weak due to the shortness of time. We still need to interpret the low R2 value. Linear regression models are described by the equation of the regression line. The fit statistic of the model shows the applicability of the model, and the value of R2 shows the limits of applicability. How can we interpret the phenomenon that the model is highly significant, but the explanatory power is low? The solution can be found in Table 2, in the 95% confidence interval columns. Wide intervals indicate that the relationship found exists, but the data scatter around the regression line is large. Since the regression coefficients and fitted values are significantly related to the average, our model can be used. A low $R^2$ value only indicates that the uncertainty of the predictions based on the model is greater than the uncertainty hoped for when the model was created. To remedy this, additional variables may be examined in a subsequent research step. However, a low $R^2$ does not reduce the importance of any significant variable. Statistically significant *p*-values still identify relationships and are coefficients also have the same interpretation. In general, therefore, there is no further reason to ignore these findings.

**Table 2.** The role and effect of year and status on financial attitudes ($R^2$ = 0.0256, *p* < 0.001) (Source: Authors' own elaboration).

| Predictor | Estimate | SE | 95% Confidence Interval | | *p* |
| --- | --- | --- | --- | --- | --- |
| | | | Lower | Upper | |
| Intercept | −71.039 | 16.289 | −102.98 | −39.100 | <0.001 *** |
| Year of survey (Y) | 0.036 | 0.008 | 0.02 | 0.054 | 0.039 ** |
| Work Status of the student (S) | −0.066 | 0.007 | −0.08 | −0.054 | <0.001 *** |

**, *** = level of significance.

Comparing the marginal averages of the groups created based on the explanatory variables of the model, we can see that the students with only university status have the highest financial attitude, followed by those who do physical work in addition to their studies, and finally those who also do intellectual work in addition to being a university student. The trends are almost completely the same in the two years of the study, therefore the interaction variable between the status and the year of the survey was not significant either (Table 2 and Figure 3).

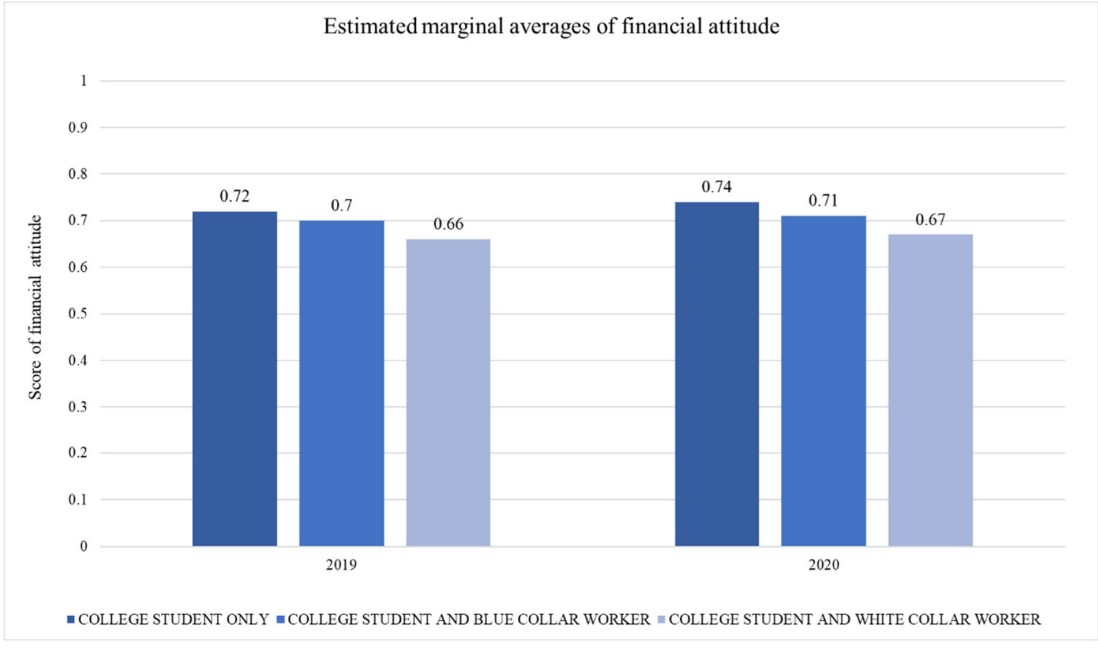

**Figure 3.** Estimated marginal averages of financial attitude at the two survey times according to occupational category (Source: Authors' own elaboration).

When comparing the marginal averages of the groups formed based on the three statuses, there was a significant difference only between those doing intellectual work in addition to the university and those who were only university students in both years and in the entire sample, in the case of the other categories in favour of the latter. In addition, in the entire sample, a significant difference could be detected between those who do physical work in addition to their studies, and those who do intellectual work, on the basis of which it can be proven that those who do physical work in addition to the university have a higher financial attitude compared to those who do intellectual work (Table 3).

**Table 3.** Estimated marginal averages of financial attitude at the two survey times according to occupational category (Source: Authors' own elaboration).

| Year | Occupational Category | College Student without a Job | College Student and Blue-Collar Worker | College Student and White-Collar Worker |
|------|----------------------|-------------------------------|----------------------------------------|------------------------------------------|
|  | **Group Average** | **0.73** | **0.71** | **0.67** |
| 2019 | 0.69 | 0.72 | 0.70 | 0.66 |
| 2020 | 0.71 | 0.74 | 0.71 | 0.67 |

If we compare the results of two years, we can also show the difference for the whole sample, which means an increase, and we also experienced a significant increase between university students and those who do physical work in addition to the student status from 2019 to 2020 (Tables 3–5).

**Table 4.** Significance values of differences between estimated marginal averages of financial attitude at two different survey times according to occupational category (Source: Authors' own elaboration).

| Occupational Category | In a Body | 2019 | 2020 |
|-----------------------|-----------|------|------|
| College student and blue-collar worker–College student and white-collar worker | 0.011 | 0.102 | 0.107 |
| College student and blue-collar worker–College student only | 0.148 | 0.171 | 0.685 |
| College student and white-collar worker–College student only | <0.001 | <0.001 | <0.001 |

**Table 5.** Significance values of differences between estimated marginal averages of financial attitude at two survey times according to discipline (Source: Authors' own elaboration).

| Year | In a Body | College Student Only | College Student and Blue-Collar Worker | College Student and White-Collar Worker |
|------|-----------|----------------------|----------------------------------------|------------------------------------------|
| 2019–2020 | 0.009 | <0.001 | 0.020 | 0.208 |

### 3.2. Investigation Knowledge

Students who work alongside the university earn money, which they typically use to meet their own wants and needs. Therefore, it does not matter how efficiently they use their income. It is therefore understandable that their interest in financial matters is at a higher level than that of their peers who are financially supported by their parents, while they only have to concentrate on their studies. One possible way to use income effectively is to secure your financial future by creating your own personal financial safety net. This safety net brings together the financial resources and opportunities that ensure students even in the event of financial difficulties caused by an unexpected crisis. This safety net can be divided into a short-term (savings) and a longer-term (investments) part. Both parts require special financial awareness. Among these, our present work examined the evolution of investment-related knowledge.

Among the five questions examining investment knowledge, described in the Methodology chapter, we attributed a key role to question Q4 (There are risk-free investments). This can be used as a kind of quick test to assess investment knowledge. The results show that only 65.54 percent of respondents answered this question correctly. Among students majoring in economics, this rate is 80.64 percent. Among the students not majoring in economics, the law students should be highlighted, with their 82.87 percent correct answer rate, the ratio among other students was only 57.81 percent (Figure 4).

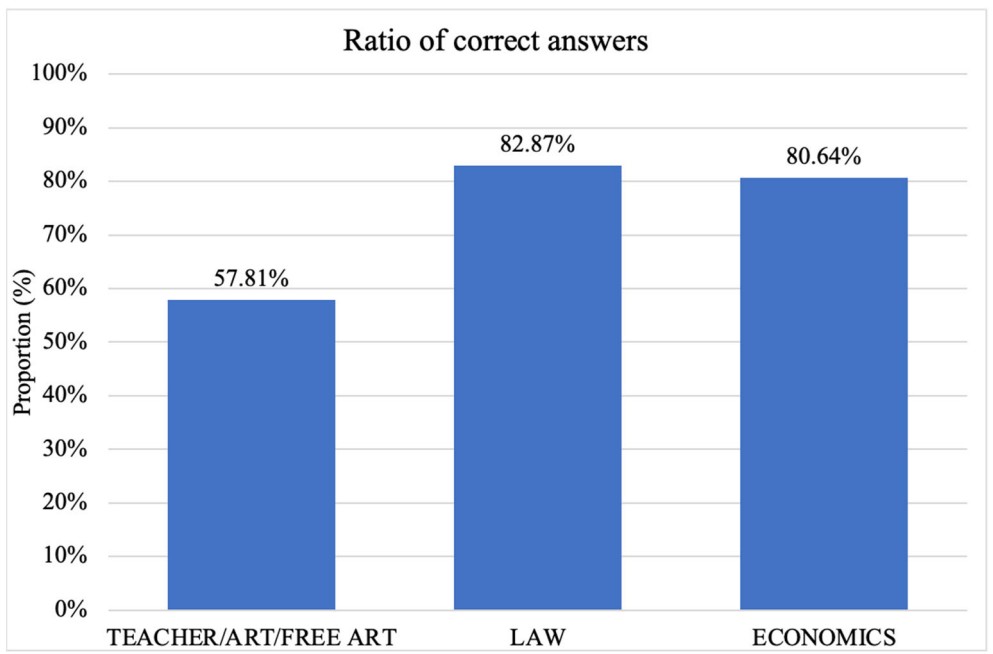

**Figure 4.** Ratio of correct answer to the statement "there are risk-free investments" (Source: Authors' own elaboration).

Based on the examination of the averages of the answers to the questions, we found that the answers to the first three questions improved during the second survey, the performance of the students worsened in the fourth, while it did not change in the fifth. We highlighted this fourth question in the previous ones, drawing attention to the fact that the mistaken belief in the existence of risk-free investments can be the cause of many financial failures. We found a significant difference when examined by major. Students of economics (and law) majors performed higher in all investment questions compared to their peers studying in other majors (Figure 5). The demonstrable improvement in the proportion of correct answers to the third question (Q3—"Money market funds have no risk, as our money is only invested in bank accounts and securities") clearly indicates progress in the field of theoretical knowledge. The most striking difference between students majoring in economics and law appears in the fourth question, which has already been mentioned several times. The proportion of those giving correct answers decreased in the former group, and increased among the latter. Based on our results, it seems that economics students who are considered realistic in ordinary" everyday life are more inclined to take even unreasonable risks in a crisis. One of the possible reasons for this is that they are overly confident in their acquired knowledge. Financiers became similarly careless before the 2008 crisis, one of the consequences of which was excessive lending, which eventually led to the outbreak of the crisis.

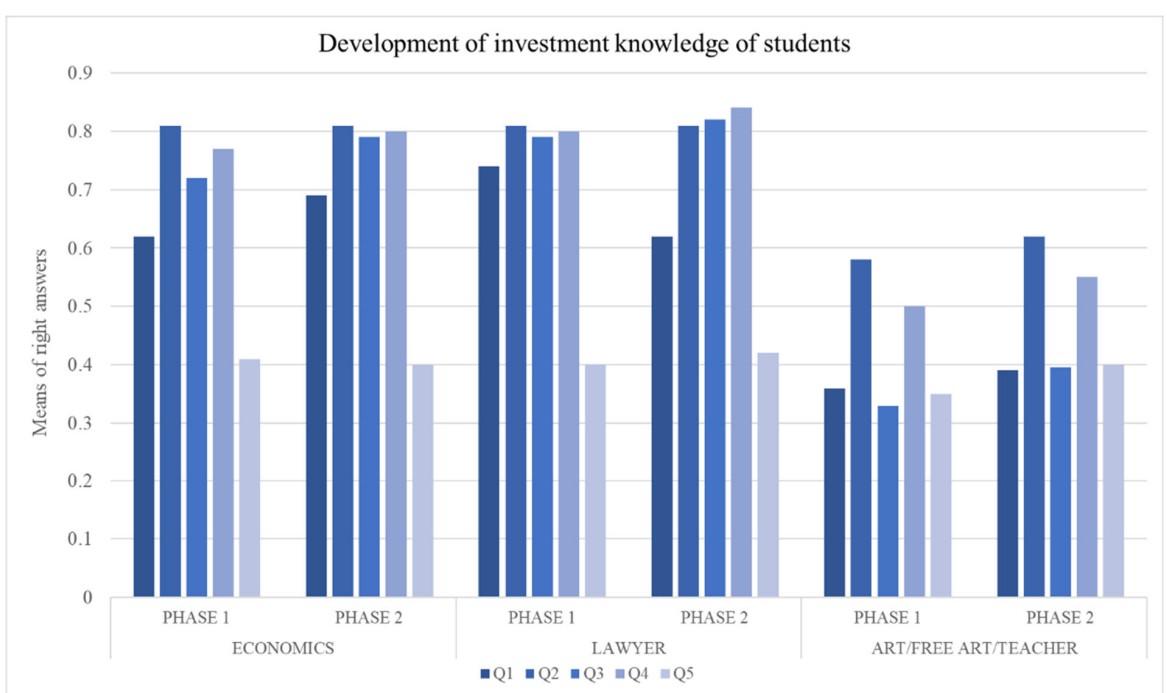

**Figure 5.** Development of investment knowledge of students specialized in disciplines within and outside the field of economics (Source: Authors' own elaboration).

In the case of economics students, an additional question arose as to what extent their knowledge can be attributed to the influence of education and how much role practice plays in it. For this reason, we compared the performance of students who only studied and those who took on a job in addition to their studies (Figure 6). During both inquiries, the performance of the students who also took on work proved to be significantly higher than that of their peers who only studied. This indicates that during their active presence and activities in the labour market, they also gain practical experience that full-time students who only study their studies do not have.

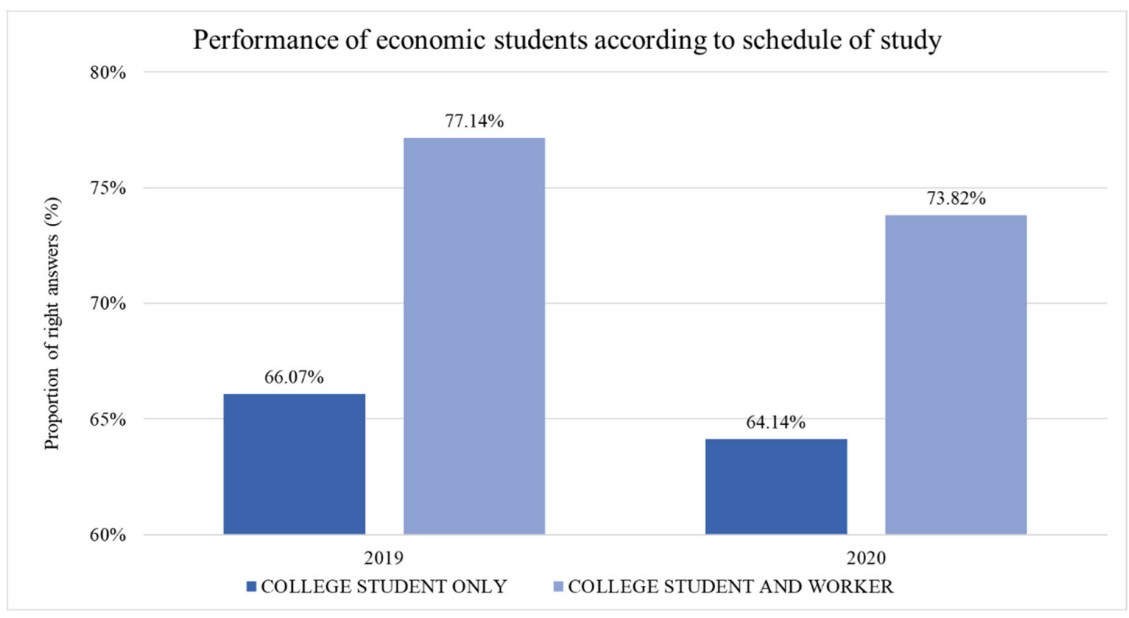

**Figure 6.** Performance of students by specialty of economic according to schedule of study (Source: Authors' own elaboration).

## 4. Discussion

Based on our model, we proved that the level of financial attitude increased significantly from 2019 to 2020 in the case of only university students and those who do physical work alongside the university. This result is favourable because we found opposite conclusions in the literature, which showed an increase in anxiety [51]. One possible explanation for our result is that a crisis situation was a motivation for them. Therefore, they turned more intensively than usual to financial affairs in order to prevent their financial situation from deteriorating as much as possible. The role of attitude is important because, according to the literature, it affects financial well-being to the same extent as actual financial behaviour [52]. The increase was also detectable for the entire sample. By comparing university students with different statuses, we were able to set up an attitude-based ranking from the lowest to the highest attitude: intellectual work alongside university–physical work alongside university–only university studies. At the two ends of the scale, we found a significant difference in the level of financial attitude between only university students and those who do manual work in addition to studying, in favour of only university students. The financial attitude of those who are in the middle of the ranking, who do intellectual work alongside the university, is only significantly higher in the entire sample compared to those who do intellectual work alongside the university. These results are in line with the finding that university education typically affects the attitudes of white-collar workers, but not at all for blue-collar workers [53]. The role of practice in determining financial knowledge and attitude is confirmed by the fact that the correlation has already been proven in other scientific fields [54].

It is a surprising result that the performance of the law students even surpassed that of the economic training participants. One possible explanation for this is that they were able to answer better based on their high level of economic law knowledge. The fact of similarity between the performance of economic and law students does occur in the literature, but it is quite rare. We found only one previously mentioned study [48] that reported this result. These authors interpreted their results in their work as meaning that the financial culture of economics students is not adequate, since law students also performed similarly in the study. On the other hand, we rather found that the financial literacy of students studying economics is significantly higher than that of students studying other subjects–with the exception of law students, who outperform their peers studying other subjects with a similarly high level of performance.

As a result of the 2020 crisis, the performance of full-time students dropped by almost two percentage points. One of the possible reasons for this is the transition to online education–partly due to a change in the efficiency of education, partly due to a decrease in student participation activity. Those students who also worked, at the same time, achieved a similar change but in the opposite direction, i.e., an increase in knowledge level of almost two percentage points. The fact that students working alongside their university studies performed better in our study is in line with observations in the literature [55], and in fact, such an effect of the work undertaken and carried out in addition to studying can already be seen in the secondary school age group [56]. The positive impact of the pandemic as a crisis on the financial culture of university students has also been confirmed by other studies [57].

According to financial knowledge, we can say the following. We were only able to partially verify our H1 hypothesis—in addition to economics students, law students also perform better than their peers who attend other courses. Our H2 hypothesis about the importance of practice in acquiring knowledge was fully confirmed.

Based on our results, we can say that we managed to verify our hypothesis H3, according to which the lack of practical experience increases university students' interest in finance. However, we found a significant difference only between students who only studied and students who did intellectual work in addition to the university.



## 5. Conclusions

In this study, we examined the financial attitudes and investment knowledge of Hungarian university students studying economics and non-economics. On the one hand, we were interested in whether the students get significantly more and more useful information during economic studies than students of other majors. On the other hand, we also investigated whether interest in financial matters improves or becomes more intense as a result of crises. During the analysis of our results, we managed to verify our hypotheses, which we formulated after reviewing the literature. These results are consistent with findings from other studies. How did you succeed in enriching the existing image with new information? That is what we are going to talk about now.

We give the following answer to the main question of our article, which was formulated in the title: our results proved that the financial attitude of university students depends on the one hand on the specialization pursued by the student, and on the other hand–to an extent that also depends on the nature of the training—Increases as a result of crises. We also managed to demonstrate an increase in the amount and quality of financial knowledge in crises, primarily among students of economics and law majors. It has also been proven that the practical experience gained in connection with the work carried out alongside the university complements the financial knowledge.

However, the answers to the investment questions proved that the fear of financial losses, even with the right knowledge, predisposes one to take more risks than necessary. And this can be the basis of bad investment decisions and can further worsen the financial situation already shaken by the crisis. The increased tendency to take risks is equally characteristic of students of all majors, which indicates that theoretical knowledge is not enough to handle a crisis. There is a need for practical experiences, the results of which can be used during the next crises. But even the combination of theory and practice is not enough if it is not accompanied by a confidence that does not waver, no matter how serious the problem that arises. Acquiring this takes time. Therefore, financial education should be started as early as possible. Time is an advantage for children that should not be wasted. Our complex experiences draw attention to the fact that ensuring the sustainability and crisis resistance of personal finances requires further development of financial education.

The present study presented a small part of a European regional survey conducted by the authors. Its novelty is primarily due to the fact that regional experiences can contribute valuable information to solving the problems of our globalizing world. The authors also wanted to achieve this goal with their original research.

**Author Contributions:** Funding acquisition, Z.Z; Resources, Z.Z.; Conceptualization, Z.Z., Methodology, B.G.K.; Investigation, B.G.K.; Visualization, J.B.; Data curation, J.B.; Supervision, L.P.; Writing—review & editing, L.P. All authors have read and agreed to the published version of the manuscript.

**Funding:** This research received no external funding.

**Institutional Review Board Statement:** The study was conducted in accordance with the Code of Human Research Ethics Published by The British Psychological Society (St Andrews House, 48 Princess Road East, Leicester LE1 7DR). Ethical review and approval were waived for this study due to the following reason: the research was conducted using voluntary and anonymous online questionnaires that did not ask for any personal information.

**Informed Consent Statement:** Not applicable.

**Data Availability Statement:** Not applicable.

**Conflicts of Interest:** The authors declare no conflict of interest. The funders had no role in the design of the study; in the collection, analyses, or interpretation of data; in the writing of the manuscript; or in the decision to publish the results.

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
