# Peer review of "The Evolution of University Students’ Financial Attitudes and Their Role in the Sustainability of Personal Finances"

_sustainability, doi:10.3390/su15086385_

Round 1

Reviewer 1 Report (Previous Reviewer 2)

Hypothesis H1 seems to be unfinished - please check it.

Author Response

  1. Hypothesis H1 remained in the article from an older version, we deleted it because we examined a total of 3 hypotheses and decided on that many in the discussion.. (Kindly see also at reviewer #3 comment #2.)

Reviewer 2 Report (Previous Reviewer 3)

Remark 1: Elaborations made by the authors sufficiently.

Remark 2: Justification is adequate.

Remark 3: Concerns associated with Remark 3 are alleviated and changes

seem to be made by the authors.

Remark 4: Justification is clear and sufficient.

Remark 5: Concerns regarding Remark 5 are addressed and articulated by

the authors.

Remark 6: Explanations are sufficient.

Remark 7: Fully addressed.

Remark 8: Hypotheses are now explicitly incorporated as requested, as

they ought to be.

Author Response

  1. Thank you.
  2. Thank you.
  3. Thank you.
  4. Thank you.
  5. Thank you.
  6. Thank you.
  7. Thank you.
  8. Thank you.

Reviewer 3 Report (New Reviewer)

Dear authors,

Thank you very much for the opportunity to read and comment on your research and manuscript.

Please see below my comments:

              Overall comment: Improvement on financial literacy should not be a task for Universities where students specialise, but should be considered in secondary schools. Havings said this, the question arises of how valuable research is that looks into University eductated young adults from different study backgrounds.

1)      Introduction:

Please explain in more detail the relevance of the paragraph (line70 to 78).

2)      Hypotheses H1 is incomplete. In my copy it reads: “deposit and foreign currency accounts,”

General comment: The value of H2 and in consequences the relates research questions is very low. Why would you not expect that students in a study program that includes financial topics perform better on financial literarcy and performance than students who do not have finance-related study topics?

Your manuscript looks at students’ financial attitudes (and financial literacy) but your research seems to be limited to investments.

3)      Methods:

Please check the English language. Q1 in its current form is not correct: “With a morgatgage combined with a life insurance…..). In Q3 your write”…money is invested in bank accounts….” There are no investments in bank accounts – please clarify the English language and please be more precise in your translation.

Please explain the choice of research questions. While you indicate in your introduction that savings are important for students, none of your research questions targets knowledge on savings, etc. For me, the questions are very random and are not a good choice to test financial literacy related to common needs. With common needs I mean the need of financial knowledge of an average adult that will be confronted with basic savings and common loans/credits. I doubt that the knowledge about money market funds can be considered as such needed basic knowledge. If your research intended to study higher levels of financial literacy, then please state this in your manuscript.

Additionally, a survey based on five survey questions (especially the five questions you chose) is not sufficient to valuate financial attitutes.

You conducted two surveys, one before the pandamic and one during the pandemic. As you want to analyse the impact of the pandemic on financial literacy, you should investigate how students were impacted by the pandemic and to which extent does this impact financial literacy. Europe slowly felt the impact of the pandemic during 2020. It therefore is important to consider the exact time of the survey and the real impact on students’ lives at that time.

Please explain how you treated answer 2,3,4 on your 5 point likert scale.

4)      In lines 327-329 you defined “employment status” as indicator whether or not students are committed to their university studies. This is wrong. Students who are fully committed but due to their financial circumstances work CANNOT be considered not to be committed. Equally, students might take work responsibilities in their study field to deepen and expand their knowledge which would again be an indicator for full commitment.

Your catogrorisations in lines 333-334 are not suffient.

5)      As you consider Q4 as the major question related to investment knowledge, please describe on your 5 point linkert scale, what you considered as a correct answer.

6)      Discussion:

Your arguments in your discussion can be improved. As we know from behavioural sciences especially an increase in anxiety (line 479) can lead to the wish for more security which leads to less spending and more savings. At the same time people might feel the need to be more vigilant and gain knowledge on how to save money as well as how to make money.

Overall, setting the above mentioned aside, your manuscript needs to highlight the research gap and your contribution to the field. As you correctly stated, previous researches covered most of your work. Please highlight the novelty of your work. As I stated at the beginning the solution to the problem is not on University level but on secondary school level.

Kind regards

Author Response

Reviewer report #3

Reviewer’s comments and suggestions

  1. Improvement on financial literacy should not be a task for Universities where students specialise, but should be considered in secondary schools. Having said this, the question arises of how valuable research is that looks into University educated young adults from different study backgrounds.
  2. Introduction
    1. Please explain in more detail the relevance of the paragraph (line70 to 78).
  3. Hypotheses H1 is incomplete. In my copy it reads: “deposit and foreign currency accounts,”
    1. General comment: The value of…
      1. …H2 and
      2. in consequences the relates research questions…

…is very low. Why would you not expect that students in a study program that includes financial topics perform better on financial literacy and performance than students who do not have finance-related study topics?

  1. Your manuscript looks at students’ financial attitudes (and financial literacy) but your research seems to be limited to investments.
  1. Methods
    1. Please check the English language.
      1. Q1 in its current form is not correct: “With a mortgage combined with a life insurance…..).
      2. In Q3 your write”…money is invested in bank accounts….” There are no investments in bank accounts –

please clarify the English language and please be more precise in your translation.

  1. Please explain the choice of research questions. While you indicate in your introduction that savings are important for students, none of your research questions targets knowledge on savings, etc. For me, the questions are very random and are not a good choice to test financial literacy related to common needs. With common needs I mean the need of financial knowledge of an average adult that will be confronted with basic savings and common loans/credits. I doubt that the knowledge about money market funds can be considered as such needed basic knowledge. If your research intended to study higher levels of financial literacy, then please state this in your manuscript.
  2. Additionally, a survey based on five survey questions (especially the five questions you chose) is not sufficient to valuate financial attitudes.
    1. You conducted two surveys, one before the pandemic and one during the pandemic. As you want to analyse the impact of the pandemic on financial literacy, you should investigate how students were impacted by the pandemic and to which extent does this impact financial literacy. Europe slowly felt the impact of the pandemic during 2020. It therefore is important to consider the exact time of the survey and the real impact on students’ lives at that time.
    2. Please explain how you treated answer 2,3,4 on your 5 point likert scale.
    1. In lines 327-329 you defined “employment status” as indicator whether or not students are committed to their university studies. This is wrong. Students who are fully committed but due to their financial circumstances work CANNOT be considered not to be committed. Equally, students might take work responsibilities in their study field to deepen and expand their knowledge which would again be an indicator for full commitment.
    2. Your categorisations in lines 333-334 are not sufficient.
  3. As you consider Q4 as the major question related to investment knowledge, please describe on your 5 point Likert scale, what you considered as a correct answer.
  4. Discussion
    1. Your arguments in your discussion can be improved. As we know from behavioural sciences especially an increase in anxiety (line 479) can lead to the wish for more security which leads to less spending and more savings. At the same time people might feel the need to be more vigilant and gain knowledge on how to save money as well as how to make money.
    2. Overall, setting the above mentioned aside, your manuscript needs to highlight the research gap and your contribution to the field. As you correctly stated, previous researches covered most of your work. Please highlight the novelty of your work. As I stated at the beginning the solution to the problem is not on University level but on secondary school level.

Co-authors’ reply (1 by 1)

  1. We have added a section to the Introduction chapter (around line 40) that explains why we examined university students and not high school students
    1. The justification was done by inserting a text
  2. Hypothesis H1 remained in the article from an older version, we deleted it because we examined a total of 3 hypotheses and decided on that many in the discussion
      1. This is now – after deleting the remaining H1 from the old version – already H1. At first reading, it is really evident that economics students should (should) perform better than students of other majors. Therefore, even before submission, we thought that maybe there was no point in investigating. However, according to the retrospective results, law students also perform in the same way as students on economics courses. This means that even an apparently self-evident statement is not always automatically proven. That is why we left this hypothesis in the submitted version.
      2. The questions Q1-Q5 in the article are not research questions, but questions of the questionnaire we used, the answers to which were examined in this article. The text was indeed ambiguous, so we modified it and clarified it.
    1. 52–74. we explained the reason for the narrowed focus in several lines. Our original research was really focused on financial literacy as a whole. Out of the entire sample, in this study we only focused on investment issues, as they lay the foundation for a secure future and also indicate the ability to think long-term.
      1. "The yield of life insurance combined with an investment always covers the amount to be repaid"
      2. In fact, we are talking about the money in the bank account and earning interest on it.
    1. In the answer to question 2), we indicated that these are not research questions, but questions about investment knowledge in the questionnaire we use. Our original complete research examined the financial literacy of university students majoring in economics and compared it to the knowledge of students in other majors. That is why we included in the research questionnaire investment questions that measure a higher than average level of knowledge. This is how we made a difference in the level of existing investment knowledge between the examined groups of university students. Given that the questionnaire was not originally prepared for the purpose of writing this study, we were only able to select those related to investments from among the questions included in it.
    2. We wrote about it in lines 52-74 that the 5 questions included in this study are 5 questions of a questionnaire originally used in a large questionnaire research. The 5 questions that measured investment knowledge in the entire original questionnaire. The original questionnaire was used to assess the financial literacy of university students, so they do not ask about the level of investment knowledge of the average person, but about the knowledge of university students. In the complete questionnaire, there was a separate group of questions to examine the attitude. We then calculated an attitude index from this, and now we have taken its values ready for the research that is the subject of this study. We wrote a short supplement about this in paragraph 5 of the methodology section.
      1. We supplemented the text around line 260. In it, we detailed the time of the inquiries and the situation and well-being of the university students.
      2. In the part of the methodology about the Likert scale, we included a short description and literature references, based on which we did not label the 2-3-4 values, as well as why we decided to do so.
      1. Thanks for the comment. According to the authors' intention, working alongside the university does not measure commitment to learning. It is an important feature because the student who works and has an income is already a labour market participant. Therefore, you have to make decisions about your own financial situation. So, in addition to theoretical knowledge, he also has real practical experience. We have modified the text accordingly.
      2. A student and manual worker is a university student who undertakes physical work in addition to his studies, and a student and intellectual worker is a student who performs intellectual work in addition to his studies. A student only is a student who does not work while completing his university studies. I also wrote it in the text.
    3. The question was asked as follows: To what extent do you think the following statement is true: There are also risk-free investments? For the answer, we used a Likert scale labelled only at the end points. Here, the value 1 was not true at all, and the value 5 was completely true. We did not label options 2, 3, 4, here we left it up to the respondents to decide for themselves. in the continuum between the two endpoints. We evaluated answer 1 as a good answer, all other marks were classified as incorrect answers. I also put these sentences in the text.
      1. The interpretation of our results related to anxiety and a possible explanation have been included in the text.
      2. I have added an addendum to the text before the hypotheses that answers these questions. I described the novelty of our work in the conclusion. I have already answered this in connection with the overall comment.

Round 2

Reviewer 3 Report (New Reviewer)

Dear authors,

Thank you for your comments. In the version that has been uploaded some changes that you present in the text have not been implemented in the manuscript.  Q1 for example stands unchanged, while you offered a perfectly sensible alternative: "The yield of life insurance combined with an investment always covers the amount to be repaid". As well your Q3 did not change, although investments into bank accounts do not exist. One can keep money in a bank account but not invest into an account. My might mean overnight deposits, or something similar. The problem is your English translation. Please check as well line 399.

Please make sure that all your commented changes are implemented in your manuscript.

Author Response

Thank you,

Changes have been made as requested

This manuscript is a resubmission of an earlier submission. The following is a list of the peer review reports and author responses from that submission.

Round 1

Reviewer 1 Report

Based on my perspective, I do not think this manuscript is related to a field of sustainability and sustainable education.

The topic is more related to education science or education and society.

Overall, the manscript still contains several weaknesses and substantial revisions are required to meet the standard to academic journals.

I have following comments.

The abstract is not well constructed, and hard to understand why and how this study was carried out. Please clearly indicate when the data were collected, and number of participants.

The first paragraph of the introduction was not understandable at all, and cannot present the significance of this study. It should be written in an academic form. It is better to discuss why individuals’ acquisition of financial knowledge and possession of financial attitudes are important, and to discuss how the crisis event could affect students' knlwedge and attituddes. Most importantly, the inroduction should clearly provide reasons why this needs to be explored, and what is research gap. More existing literatures need to be reviewed. The discussion on the possible effect of crisis events on possestion of finanacial knowledge and enhancement of attitudes should be explained based on relevant theories and concepts.

Research objectives are not so clear as well.

For the research method, it is hard to understand. I am also not convinced with the measurement of financial literacy and attitudes. Validity and reliability of measurement are very important and need to be analyzed. Sources of used measurement should be addressed.

The analysis method was not clearly explained in relevant to the scope of this study.

What is the attitude model.?

R2 is too low, only 2% of expnatory power.

For the statististic tests, more statistic analyses are needed to prove the significance difference between groups.

Discussion part should be imporved. Recent litereratures should be exployed to support the results. Theretical and practical contributions of this study should be claerly explained.

Reviewer 2 Report

The work is exciting but needs significant improvements to be published. These mainly concern the method used, although doubts have also been raised regarding hypotheses and research questions.

The hypotheses-research questions posed by the authors are inappropriate. In principle, only the first one is acceptable, the rest sound like survey questions or are trivial (Q4).

There is a lack of description of the variables, especially the explanatory variable in the model (financial attitude). This is of particular importance due to the assumptions of the method used in the paper (ANCOVA). 

Firstly, the method requires an examination of the distributions of the variables, which was not done. 

Secondly, the method requires an examination of homogeneity of variance (Brown-Forsythe test or Levin test).

Thirdly, the presentation of the results of the analysis needs to be refined (lack of tabularized detailed regression results to evaluate the research carried out).

In the theoretical part, some space should be devoted to the analytical method that was used (ANCOVA) and the formal notation of the model (1) should be improved - in this form, it is not very readable.

Finally, the title of Fig.1 is misleading - I think it should be the structure of the sample. (As I suppose)

Reviewer 3 Report

The paper, as explicitly stated, aims to examine how crises influence the decision-making of Hungarian university students. I herein elaborate on my concerns below and hope they help authors strengthen their manuscripts on this timely topic.

1.     To measure a given student’s investment knowledge, the authors designed five questions and contended that 4th question is the one with prime importance. Why do students know as to the existence of some risk-free investment alternatives is that important? According to whom? Is it merely a contention on the part of the authors? Elaboration is needed herein. [203-213]

2.     Also, are the select five questions sufficient to gauge the degree of investment knowledge on the part of respondent students? [203-213]

3.     The authors categorized “occupations/specializations” into three groups: economics students, law students, and the rest. What is that categorization based upon? [227-240]

4.     The method which the authors have utilized to create so-called “indexes” seems to be built upon their contention, which it should not. Why was that particular method used to create the index chosen? What is the justification? [243-253]

5.     The authors claimed that the regression analysis conducted takes into account the normality assumption. Which normality test is carried out? Where do results lie in the main text? [254-256]

6.     Are employment status and year in which respective surveys were carried out only regressors that are supposed to explain the respondent students’ “financial attitude”? What are the other possible regressors that might explain students’ financial attitudes? Which theory is linking these variables with the dependent variable? [254-260]

7.     Regression results as to model 1 seem to be explained but missing as the table in the main text. It would be better and informative had the authors presented these results in a table. [293-299]

8.     In the discussion part, the authors have depicted analysis results with attention to detail and given evidence as to whether hypotheses subject to testing are verified or not. Yet, they failed to previously write down their hypotheses in the main text. So, they appear nowhere, except at the very end of the paper wherein based on analysis results authors are discussing the implications of the findings on their hypotheses. Hypotheses shall be presented explicitly in the main text. [418-425]